# Autoimmune Hepatitis—Challenging Diagnosis

**DOI:** 10.3390/medicina58070896

**Published:** 2022-07-04

**Authors:** Aleksandra Mroskowiak, Agata Suleja, Maria Stec, Wiktoria Kuczmik, Maciej Migacz, Michał Holecki

**Affiliations:** 1Student Scientific Society at the Department of Internal, Autoimmune and Metabolic Diseases, School of Medicine, Medical University of Silesia, 40-055 Katowice, Poland; agatasuleja@gmail.com (A.S.); mariaannastec@gmail.com (M.S.); wikikuczmik@interia.pl (W.K.); 2Department of Internal, Autoimmune and Metabolic Diseases, School of Medicine, Medical University of Silesia, 40-055 Katowice, Poland; maciek.migacz@gmail.com

**Keywords:** Autoimmune Hepatitis, AIH, antibody, immunoglobulin, steroids

## Abstract

The incidence of Autoimmune Hepatitis (AIH) increases worldwide. If undiagnosed, it may progress end-stage liver disease. Unfortunately, there is no characteristic clinical presentation of this disease, which makes the illness hard to recognize. A case report illustrates the difficulties of diagnosing the patient during his two hospitalizations and his final treatment with prednisolone which improved the patient’s condition.

## 1. Introduction

Autoimmune Hepatitis (AIH), whose prevalence is increasing worldwide, is known as a generally progressive, chronic disease, which can be fluctuating. It occurs both in children and adults, but the cause ultimately remains unknown [1]. Suspected causes are viral infections, genetic susceptibility, and concomitance of other autoimmune diseases [2]. We distinguish between two types of AIH. Type 1 is more common in adults, especially in women (4:1 ratio), and is associated with elevated levels of antinuclear antibody (ANA), smooth muscle antibody (SMA), anti-actin actin antibodies (AAA). Type 2 occurs mostly in children, presents worse clinical course, and requires long-term care [3]. For type 2, we have observed the prevalence of anti-liver/kidney microsome type 1 antibody (LKM-1) and anti-liver cytosol antibody (ALC-1) [4]. Additionally, diagnosis is based on elevated aminotransferases, serum immunoglobulin G (IgG), and interface lympho-plasmacytic hepatitis. In Table 1 we present simplified diagnostic criteria for AIH. It is important to note that the clinical manifestation may differ between individuals [5]. Undiagnosed AIH may lead to cirrhosis and, eventually, end-stage liver disease. For this reason, it should not be omitted in differential diagnosis of liver impairment. Prolonged steroid therapy, despite being associated with numerous unwanted side-effects, is still estimated to be the most effective therapeutic choice in patients with moderate or severe courses [6].

## 2. Case Presentation

A 68-year-old man was admitted to the Department of Internal, Autoimmune, and Metabolic Diseases due to growing abdominal pain (VAS 4-5/10), weight loss (approximately 5 kg), jaundiced skin and whites of the eyes. Those symptoms have persisted for two weeks prior to admittance. Additionally, he had reported diarrhea (one to three times a day) occurring over the previous two months, discoloration of stool and darkening of the urine. Moreover, the patient had concomitant diseases, such as hypertension, type 2 diabetes, a history of hemorrhagic stroke (2003), chronic vascular changes in the central nervous system and sarcoidosis of larynx (since 2019). On initial physical examination patient displayed jaundice and tenderness of the abdominal cavity. The laboratory tests revealed slightly increased CRP serum level (11.1 mg/L), increased activity of ASPAT (2530 U/L), ALAT (4778 U/L), alkaline phosphatase (228 U/L), hyperbilirubinemia (23.90 mg/dL), and prolonged prothrombin time (18.1s INR 1.59). The level of copper (947 µg/L) and ceruloplasmin (24.7 mg/dL) were in normal range. During the hospitalization additional results were obtained, such as increased level of alpha1 protein (4.2%) and decreased level of alpha2 protein (7.3%), gamma protein (17.6%) were in the upper healthy range. The level of ANA IgG antibodies was scant, and the COMBI test was negative. Viral hepatitis was excluded. Microbiological tests were negative. Reactive antibodies anti-CMV IgG and IgM were confirmed and anti-EBV IgG antibodies were positive as well.

The abdominal ultrasound and CT scan showed hepatic steatosis and a polyp of the gallbladder (Appendix A, Figure A1A–F). Due to uncertainty in determining the reason for liver failure, a biopsy was performed, which revealed lymphoplasmacytic interface hepatitis (Appendix B, Figure A2a–g). During the hospitalization, steroid therapy with prendisolone was administered. Over the next few days an improvement in clinical status was observed, including the reduction of the jaundice and normalization of the liver enzymes. 

During the second admission to the Department, the patient was asymptomatic, his skin and the whites of eyes were slightly jaundiced. The laboratory tests revealed mild hyperbilirubinemia (1.95 mg/dL), slightly increased activity of ASPAT (61.3 U/L), ALAT (128 U/L), alkaline phosphatase (168 U/L), however the results were improved compared to the previous hospitalization. The comparison of the laboratory tests’ results is shown below (Table 2 and Table 3).

To avoid side effects of steroid therapy the patient was provided with azathioprine and the dose of prednisolone was decreased. As the clinical status was improved, the patient was discharged from the hospital in overall stable condition and recommended to continue the pharmacotherapy on an ambulatory basis.

## 3. Discussion

As there is no characteristic clinical presentation of AIH, the described patient may seem to suffer from acute liver failure, coinciding with jaundice and cholestasis, as the symptoms were not specific. Firstly, the inflammatory diseases of the liver were excluded [8]. The cause of AIH remains unknown, but due to the fact that it may be triggered by virus infection, previous EBV and CMV infections were suspected [9,10]. In our patient, reactive antibodies anti-CMV IgG and IgM were confirmed and anti-EBV IgG antibodies were present, which might have had an impact on the course of the disease. 

Even though the cause of AIH is still unknown, the pathophysiology is considered secondary to immune tolerance failure in a genetically susceptible individual. The immune tolerance failure is thought to lead to a T-cell mediated inflammation as a response to environmental triggers including infections, toxins and medications [11]. Regardless of pathophysiology AIH may be recognized as AIH type 1 or AIH type 2. Type 1, also known as classical type, is more common in adults. Type 2 is more often recognized in children and is associated with poorer prognosis. This classification is based on the presence of antibodies-AIH 1 is confirmed in patients positive for anti-nuclear antibodies and/or smooth muscle antibodies; AIH 2 is recognized in patients positive for anti-liver/kidney microsomal antibody type 1 and/or anti-liver cytosol antibody type 1 [12].

Several cases reported the development of AIH following Epstein-Barr virus (EBV) infection. Vento et al. showed that in 2 of 7 individuals suffering from mononucleosis AIH has developed within 4 months. Kojima et al. presented a hospitalized patient with symptoms typical of EBV infection. Even though the anti-EBV IgG was found and the circulating lymphocytes were positive for the genome of EBV, the patient was finally diagnosed with AIH based on ANAs and the liver biopsy [13]. 

CMV infection may also have an impact on AIH development. Salcedo et al. reported 60% of cases (12 out of 20 patients) with CMV suffered from AIH later on [14]. Furthermore, a large UK study, which included patients from 28 hospitals, indicates that 0.5% of patients admitted to the hospital due to AIH also had acute viral hepatitis (CMV/EBV/HEV) [15]. 

Our patient’s laboratory tests revealed significantly elevated aminotransferases (ASPAT 2530 U/L and ALAT 4778 U/L). The Level of ANA IgG antibodies was scant. However, the Combi test was negative in the described case. Moreover, positive anti-CMV IgG and IgM and anti-EBV IgG antibodies were alarming. The patient suffered from the more common type 1 AIH.

## 4. Conclusions

As the prevalence of AIH is increasing in all the patients with non-specific symptoms of the liver disease, AIH should be considered during diagnosis. Although the Simplified Diagnostic Criteria for Autoimmune Hepatitis are available, the final diagnosis is still hard to make, no clear guidance for making the diagnosis may be problematic in the medical practice, so it should be clarified and the criteria’s updating should be considered.

## Figures and Tables

**Table 1 medicina-58-00896-t001:** Simplified Diagnostic Criteria for Autoimmune Hepatitis of International Autoimmune Hepatitis Group (IAHG) [7].

Clinical Feature	Points
**ANA or SMA**	
≥1:40 ANA or SMA ≥ 1:80 or LKM1 ≥ 1:40 or SLA-positive	+1 +2
**Serum IgG**	
>upper limit of normal >1.1 times upper limit of normal	+1 +2
**Histologic findings**	
Compatible with AIH Typical of AIH	+1 +2
**Hepatitis viral markers**	
Negative	+2
**Aggregate score without treatment**	
Definite AIH Probable AIH	≥7 ≥6

ANA, antinuclear antibody; SMA, smooth muscle antibody; LKM1, liver kidney microsomal antibody; SLA, soluble liver antigen antibody; IG, immunoglobulin.

**Table 2 medicina-58-00896-t002:** The laboratory tests’ results during both hospitalizations.

Laboratory Parameters	First Hospitalization	Second Hospitalization
CRP (mg/dL)	11.1	8.5
Na^+^ (mmol/L)	130.8	136.9
K^+^ (mmol/L)	3.8	4.74
Cl^−^ (mmol/L)	96.5	101.2
Total bilirubin (mg/dL)	23.9	1.95
ALT (U/L)	4778	128
AST (U/L)	2530	61.3
GGTP (U/L)	269	300
ALP (U/L)	228	168
Amylase (U/L)	63.5	91.2
Lipase (U/L)	36.2	51.4
Total protein (g/dL)	4.91	6.44
TSH (mIU/L)	0.53	1.07
D-Dimer (ng/mL)	1005	905
INR (s)	1.59	1.07
APTT (s)	30.4	34.3

**Table 3 medicina-58-00896-t003:** The morphology results during both hospitalizations.

Morphology Parameters	First Hospitalization	Second Hospitalization
WBC (10^3^/μL)	7.43	6.52
RBC (10^6^/μL)	5.27	4.63
HGB (g/dL)	16.3	15.4
HCT (%)	44.5	43.9
MCV (fL)	84.4	94.8
MCH (pg)	30.9	33.3
MCHC (g/dL)	36.6	35.1
PLT (10^3^/μL)	293	233
RDW (%)	14.9	16.2
RDW-SD (fL)	45.6	57.0
PCT (%)	0.32	0.24
MPV (fL)	11.0	10.4
PDW (%)	12.5	11.7
P-LCR (%)	32.4	28.0
Lymphocytes (%)		27.1
Monocytes (%)		10.9
Neutrophils (%)		55.7
Eosinophils (%)		5.2
Basophils (%)		0.5

## Data Availability

Not applicable.

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
