# Peer review of "Autoimmune Hepatitis—Challenging Diagnosis"

_medicina, 2022, doi:10.3390/medicina58070896_

Round 1

Reviewer 1 Report

Case Report

 Autoimmune hepatitis - challenging diagnosis

We know that autoimmune hepatitis is liver inflammation which could be occured when the body's immune system attack liver cells.

-          This case report is one of the few cases published and it is an interesting case, but I have some comments

·         Author should first define the disease and mention the possible causes of it.

·         Usually in such case, author should do all the necessary tests to facilitate its diagnosis.

These tests include the following:-

·         Liver function tests and electrolytes, Complete blood count, Coagulation tests, and autoimmune antibodies in order to confirm the diagnosis.

·         Authors have to add the results of CBC and summarized all the biochemical and hematological tests in tables for better presentation of the results.

·         There are two major forms of AIH: type 1 and type 2, so authors should determine the type of AIH.

·         Conclusion should be corrected and add some details.

·         References should be written according to the journal instructions.

Author Response

We would like to thank you for reviewing our manuscript and thank for your valuable notes. 

  • We put in the corrected version all the results of tests that we performed and now it's in the clear table, it was very helpful note.
  • Our patient's AIH type was 1. It was and important information and it's in the corrected manuscript now. 
  • We expansed the conclusions. Our main point is there are no clear guidance for making the diagnosis and we hope other doctors will find our case helpful while making diagnosis in their patients 
  • We rewrote the references according to the journal instructions.

Reviewer 2 Report

Mroskowiak et al. presented a case of AIH, which commonly has no clinical manifestations. Here are my concerns:

(1) Appendix A and B are cited oppositely and the panels are wrong (a-g for A; a-f for B).

(2) The characteristics of lymphoplasmacytic interface hepatitis, in this case, need to be described in the main text.

(3) The authors should conclude this case report with information regarding the diagnosis and treatment of this case with some suggestions for future studies and/or doctors.

Author Response

We would like to thank you for reviewing our manuscript and thank for your valuable notes. 

  1. We rearranged the appendixes in the right way
  2. We added the characteristics of lymphoplasmacytic interface hepatitis to the main text, it was very helpful note
  3. We rewrote the conclusions and wrote the patient's ambulatory treatment as a suggestion to whom can find our case report helpful while making their own diagnosis